# A hierarchy of metabolite exchanges in metabolic models of microbial species and communities

Ylva Katarina Wedmark[1,2], Jon Olav Vik[1,2], Ove Øyås[1,2]*

**1** Faculty of Biosciences, Norwegian University of Life Sciences (NMBU), Ås, Norway, **2** Faculty of Chemistry, Biotechnology and Food Science, NMBU, Ås, Norway

* ove.oyas@nmbu.no, oveoyas@gmail.com

**Data Availability Statement:** Data and code are available at https://gitlab.com/YlvaKaW/exchange-enumeration.

**Funding:** This work was funded by the Research Council of Norway grant 248792 (DigiSal) with

## Abstract

The metabolic network of an organism can be analyzed as a constraint-based model. This analysis can be biased, optimizing an objective such as growth rate, or unbiased, aiming to describe the full feasible space of metabolic fluxes through pathway analysis or random flux sampling. In particular, pathway analysis can decompose the flux space into fundamental and formally defined metabolic pathways. Unbiased methods scale poorly with network size due to combinatorial explosion, but a promising approach to improve scalability is to focus on metabolic subnetworks, e.g., cells' metabolite exchanges with each other and the environment, rather than the full metabolic networks. Here, we applied pathway enumeration and flux sampling to metabolite exchanges in microbial species and a microbial community, using models ranging from central carbon metabolism to genome-scale and focusing on pathway definitions that allow direct targeting of subnetworks such as metabolite exchanges (elementary conversion modes, elementary flux patterns, and minimal pathways). Enumerating growth-supporting metabolite exchanges, we found that metabolite exchanges from different pathway definitions were related through a hierarchy, and we show that this hierarchical relationship between pathways holds for metabolic networks and subnetworks more generally. Metabolite exchange frequencies, defined as the fraction of pathways in which each metabolite was exchanged, were similar across pathway definitions, with a few specific exchanges explaining large differences in pathway counts. This indicates that biological interpretation of predicted metabolite exchanges is robust to the choice of pathway definition, and it suggests strategies for more scalable pathway analysis. Our results also signal wider biological implications, facilitating detailed and interpretable analysis of metabolite exchanges and other subnetworks in fields such as metabolic engineering and synthetic biology.

## Author summary

Pathway analysis of constraint-based metabolic models makes it possible to disentangle metabolism into formally defined metabolic pathways. A promising but underexplored

support from grant 248810 (Centre for Digital Life Norway). The grant was awarded to JOV. The funders had no role in study design, data collection and analysis, decision to publish, or preparation of the manuscript.

**Competing interests:** The authors have declared that no competing interests exist.

application of pathway analysis is to analyze exchanges of metabolites between cells and their environment, which could also help overcome computational challenges and allow scaling to larger systems. Here, we used four different pathway definitions to enumerate combinations of metabolite exchanges that support growth in models of microbial species and a microbial community. We found that metabolite exchanges from different pathway definitions were related to each other through a previously unknown hierarchy, and we show that this hierarchical relationship between pathways holds more generally. Moreover, the fraction of pathways in which each metabolite was exchanged turned out to be remarkably consistent across pathway definitions despite large differences in pathway counts. In summary, our work shows how pathway definitions and their metabolite exchange predictions are related to each other, and it facilitates scalable and interpretable pathway analysis with applications in fields such as metabolic engineering.

## Introduction

Metabolic pathways are combinations of biochemical reactions that occur in a cell or organism, and the interplay between pathways forms the cell or organism's metabolic network [1]. The growing availability of genomes and other omics data has enabled metabolic network reconstruction *in silico*, giving rise to genome-scale metabolic models (GEMs) that are usually formulated as constraint-based models (CBMs) to allow scaling [2, 3]. A CBM describes a metabolic network with $m$ metabolites and $n$ reactions as an $m \times n$ stoichiometric matrix in which each element is the stoichiometric coefficient of a metabolite in a reaction. By making the quasi-steady-state assumption, justified by the fact that metabolism is very fast compared to other biological processes [4], the mass balances of the metabolites can be written as

$$\mathbf{Nr} = 0, \tag{1}$$

where $\mathbf{N}$ is the stoichiometric matrix, and $\mathbf{r}$ is the vector of fluxes (reaction rates). Solving for $\mathbf{r}$ yields a flux vector that satisfies this linear system of equations, i.e., a feasible combination of reaction rates at steady state.

There are infinitely many flux vectors that satisfy Eq 1 and thus form the null space of $\mathbf{N}$. However, some of these solutions are not realistic because of physical, chemical, or environmental limitations. This can be taken into account by additional linear constraints, most commonly lower and upper bounds on fluxes:

$$r_i^{\text{lb}} \leq r_i \leq r_i^{\text{ub}}, \tag{2}$$

where $r_i^{\text{lb}}$ and $r_i^{\text{ub}}$ are the lower and upper flux bounds of reaction $i$, respectively. Geometrically, these bounds are hyperplanes that eliminate infeasible solutions by slicing the feasible flux space. In the simplest case, bounds are only applied to irreversible reactions to ensure flux in one direction, i.e., $r_i^{\text{lb}} = 0$ or $r_i^{\text{ub}} = 0$. These constraints are homogeneous, meaning that the right-hand-side is always zero. They slice the null space to a cone that can be further sliced to a more general polyhedron by adding non-zero bounds, i.e., $r_i^{\text{lb}} \leq r_i^{\text{ub}} \neq 0$ or $r_i^{\text{ub}} \geq r_i^{\text{lb}} \neq 0$, or other inhomogeneous constraints [5].

The feasible flux space of a CBM (Fig 1) can be analyzed by biased or unbiased methods [6]. Biased methods such as flux balance analysis (FBA) focus on specific flux vectors that optimize an assumed cellular objective [7]. For microbes, it can be reasonable to assume that the objective of a cell is to grow as fast as possible, i.e., to maximize growth rate as modeled by a biomass

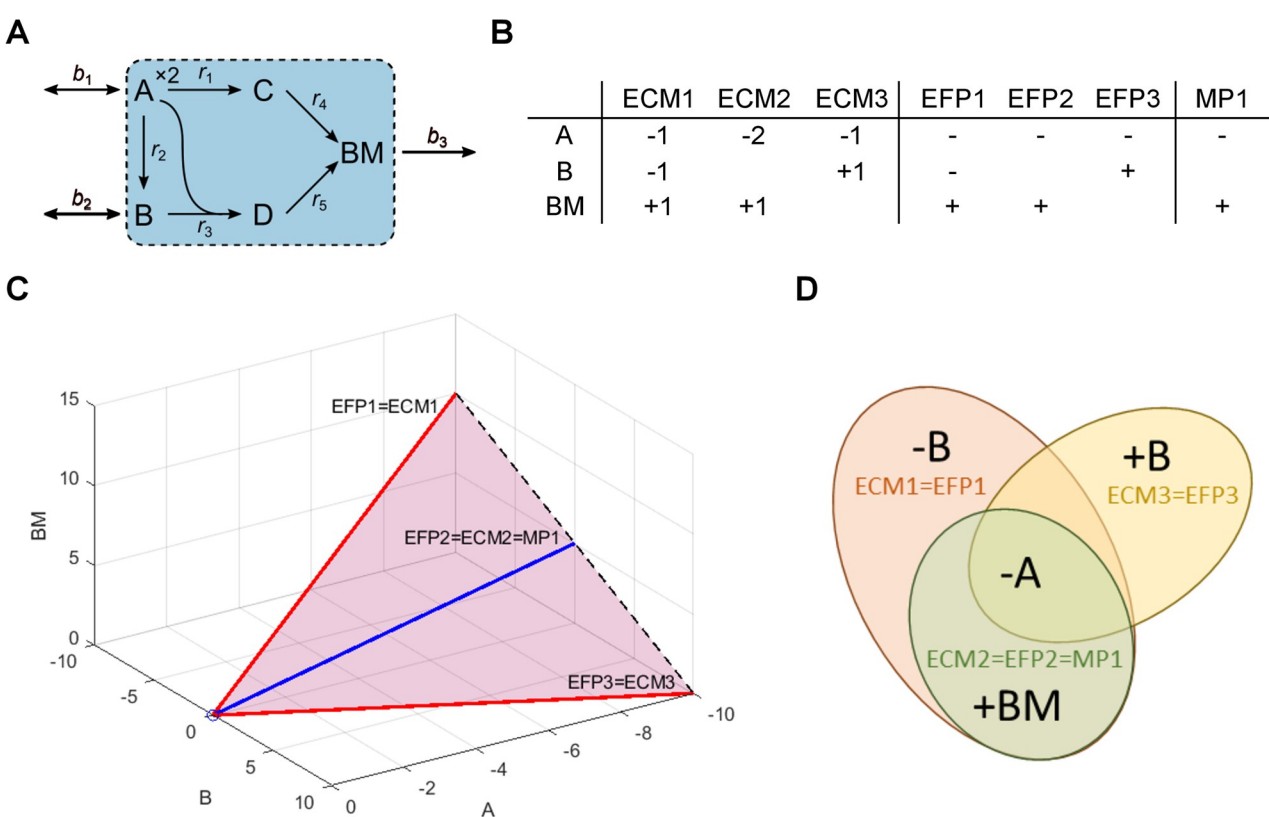

**Fig 1. Comparison of pathways for an example metabolic network.** (A) Network with five internal reactions ($r_1$–$r_5$), three boundary reactions ($b_1$–$b_3$), and five metabolites (A–D and BM). Two A are needed to produce one C in $r_1$. The external metabolites A, B, and BM are imported or exported by their respective boundary reactions. BM represents biomass and the flux of its boundary reaction is the growth rate. (B) The elementary conversion modes (ECMs), elementary flux patterns (EFPs), and minimal pathways (MPs) of the network. ECMs include stoichiometry, while EFPs and MPs are given as flux patterns (− and + representing import and export, respectively). (C) The conversion cone of the network with ECMs, EFPs, and MPs represented as rays. The ECMs generate the cone without cancellations and the EFPs and MPs correspond to the flux patterns of the ECMs. The cone is unbounded in the direction of increasing import of A and bounded by the extreme rays, ECM1 = EFP1 and ECM3 = EFP3 (red). The last ray, ECM2 = EFP2 = MP1 (blue), is the only one that corresponds to an MP because it is the only minimal set of metabolite exchanges required to produce biomass. (D) Venn diagram comparing flux pattern representations of ECMs, EFPs, and MPs. The ECMs and EFPs overlap completely in this example, but this is not true in general.

reaction, but this assumption does not hold for all conditions or cells [8]. Unbiased methods avoid such assumptions by characterizing the entire solution space through random flux sampling [9] or pathway analysis [10]. In flux sampling, Monte Carlo methods are used to draw random solutions uniformly from the feasible flux space, and the resulting samples provide flux probability distributions [11]. Pathway analysis generally involves enumeration of formally defined metabolic pathways, e.g., elementary flux modes (EFMs) or elementary flux vectors (EFVs), which are the minimal conformal generators of the flux cone and flux polyhedron, respectively [5]. This means that EFMs and EFVs are sets of vectors that can combine into all possible flux routes through a network without cancellation of fluxes. Biased methods scale to GEMs and combinations thereof, but unbiased methods are often limited to smaller CBMs due to combinatorial explosion: both the number of samples needed to uniformly cover the flux space and the number of pathways increase combinatorially with network size [12].

Uniform random flux sampling has become feasible for GEMs in recent years [13], but to scale pathway analysis it is necessary to focus on a subset of pathways or a subnetwork of the full metabolic network [14]. For example, elementary flux patterns (EFPs) have been defined as unique sign patterns (+/−) of EFMs from a full network that pass through a subnetwork, specifically the patterns that cannot be generated from other patterns without cancellations [15]. Many biological questions are primarily concerned with organisms' interactions with each other or their environment [16], and in these cases one can ignore internal reactions entirely to focus on the subnetwork of boundary reactions that allow metabolite exchange. This is the motivation for elementary conversion modes (ECMs): net metabolite conversions that, analogously to EFVs, are the minimal set of conformal generators of a conversion cone [17, 18]. Like EFPs, ECMs can be generalized to arbitrary subnetworks. In this case, the ECMs are equivalent to projected cone elementary modes (ProCEMs), a subset of which has been shown to correspond to EFPs [19]. We will refer to these pathways as ECMs, also for general subnetworks. Both EFPs and ECMs are currently limited to homogeneous constraints, but minimal pathways (MPs), defined as minimal sets of reactions in a subnetwork that must have non-zero flux to satisfy all constraints on the full network, allow arbitrary subnetworks and constraints [20]. ECMs, EFPs, and MPs can be compared as flux patterns, which, when applied to boundary reactions, represent net metabolite conversions (Fig 1).

Metabolic pathway analysis has long been used to gain new knowledge about biological networks, in particular in fields such as metabolic engineering and synthetic biology. The possibility of enumerating all the distinct modes in which a metabolic network can operate makes pathway analysis very useful for exploring the capabilities of a cell as a whole. For example, one can identify all possible routes a cell can use to convert substrates into products, which, in turn, allows design of knock-out strategies for selection of routes that couple growth to production with high productivity or yield [21, 22]. Enumerated pathways also directly reveal essential reactions, which must always have non-zero flux, and blocked reactions, which can never have non-zero flux, as well as couplings between reactions [5]. Reactions that are neither essential nor blocked can be ranked and correlated by their relative importance for growth or other network functionalities [20]. Enumerated pathways can also be used for dynamic modeling and optimization of cell factories, e.g., in hybrid cybernetic models that simulate biomass and metabolite dynamics by distributing resources optimally among pathways [23, 24]. Pathway definitions that allow direct targeting of metabolic subnetworks are particularly useful for applications in synthetic biology, where small pathway modules are typically studied in the context of global metabolism [25]. Subnetwork-based methods are also well-suited for analyzing multicellular systems such as microbial communities: by focusing on metabolite exchanges, one can study interactions between cells and design communities. However, few studies to date have applied pathway analysis to multicellular systems, likely because of scalability issues [26].

Although several unbiased methods suitable for analysis of metabolite exchanges are available, it remains unclear how they relate to each other in terms of prediction and biological interpretation of metabolite exchanges. To address this, we enumerated ECMs, EFPs, and MPs in metabolic models ranging in scope from cells to communities and in size from core models to GEMs. Focusing on flux patterns that support growth, we found that the MPs were always a subset of the EFPs, which in turn were always a subset of the ECMs. We also enumerated EFMs for all core models and found that the ECMs were always a subset of the subnetwork flux patterns from EFMs. Moreover, metabolite exchange frequencies, i.e., the fraction of enumerated pathways that included exchange of each metabolite, were mostly stable across pathway definitions, showing that the same biological conclusions can be drawn from different methods. Flux sampling scaled to all analyzed models and complemented pathways with flux

probabilities, particularly by sampling individually for each enumerated pathway. Overall, our results allow unbiased methods to be understood in conjunction with each other, which should enable users to choose the most appropriate approach for their questions.

## Results

### Metabolite exchanges in microbial species

We first applied pathway enumeration to an *E. coli* core model (e_coli_core) [27] with 95 reactions, 20 of which were boundary reactions, and an *H. pylori* GEM (iIT341) [28] with 554 reactions, 77 of which were boundary reactions (Fig 2). For e_coli_core, we enumerated 100,274 EFMs for the full network, from which we obtained 1,004 unique flux patterns in the subnetwork of boundary reactions. From these patterns, 738 of which were growth-supporting, we extracted 118 EFPs, 63 of which were growth-supporting. We also enumerated 689 ECMs, corresponding to 346 growth-supporting patterns, and 34 MPs, all of which were growth-supporting patterns by definition. The computation time for enumeration was 47 s for EFMs, 8 s for ECMs, and 2 s for MPs (S1 Fig). As demonstrated previously [19], we found that it was possible to extract the same 118 EFPs from the ECMs as from the EFMs. EFM enumeration was infeasible for iIT341 with the computational power available to us, but we enumerated 874,236 ECMs, 125,020 of which were unique and growth-supporting, as well as 1,304 MPs. ECM and MP enumeration took about 1h 9 min and 3 min, respectively (S1 Fig). Based on our observations from e_coli_core, we used the ECMs to extract 16,573 EFPs, 5,878 of which were growth-supporting. Thus, we consistently found more ECMs than EFPs and more EFPs than MPs. Enumerated pathways included the same metabolites across pathway definitions (16 for

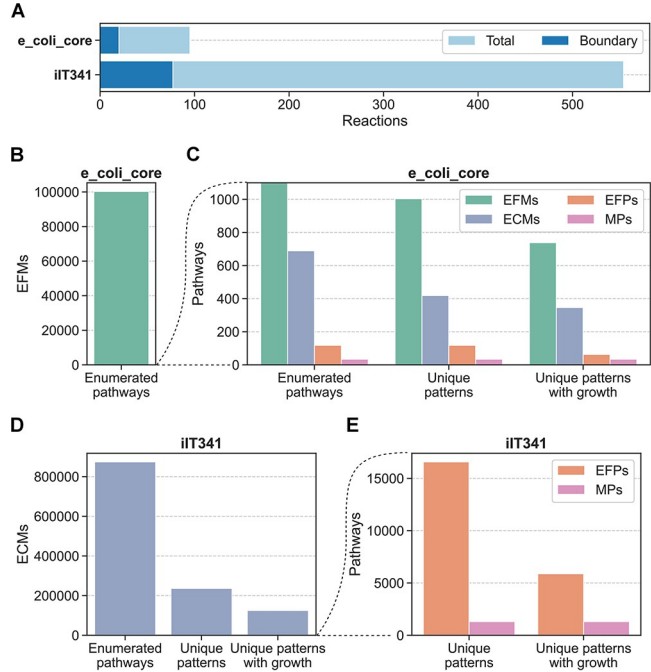

**Fig 2. Number of reactions and pathways for microbial species.** (A) Number of reactions and boundary reactions in e_coli_core and iIT341. (B) and (C) Number of pathways for e_coli_core. (D) and (E) Number of pathways for iIT341. From the full set of enumerated pathways, we extracted the unique patterns of metabolite exchanges. From these patterns, we extracted the unique patterns with growth (positive flux for the biomass reaction).

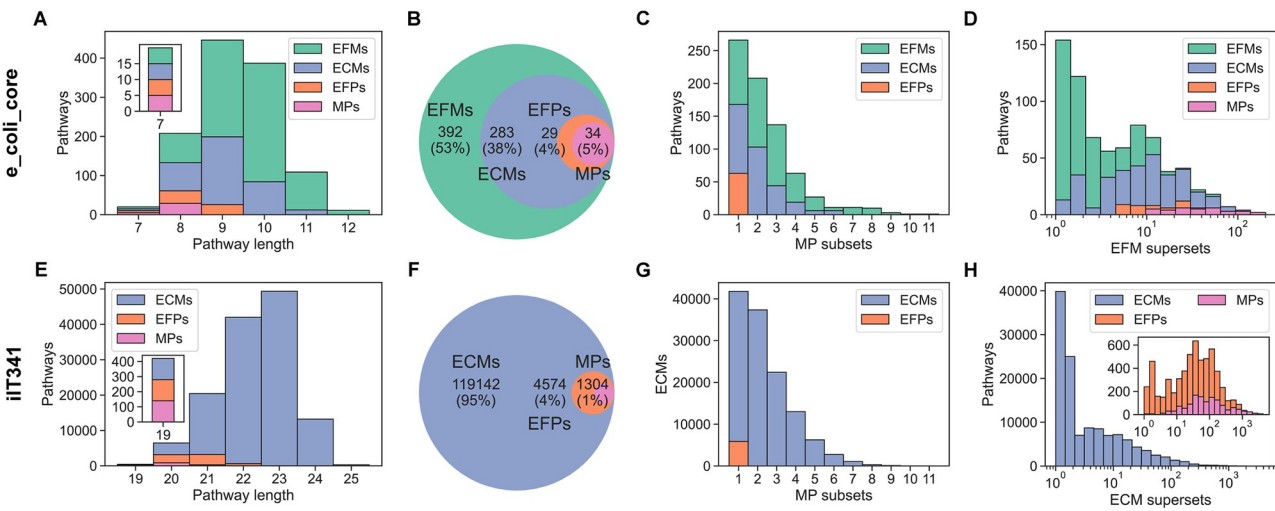

**Fig 3. Comparison of pathways for microbial species.** (A)–(D) Comparison of pathways for e_coli_core. (E)–(H) Comparison of pathways for iIT341. (A) and (E) Distribution of pathway lengths. Bars are stacked on top of each other. The insets show the shortest pathways. (B) and (F) Venn diagrams of pathways. Number of unique growth-supporting patterns and percent of the total number of patterns are shown. (C) and (G) Distribution of the number of MPs that are a subset of each of the other pathway definitions. Bars are layered behind each other. (D) and (H) Distribution of the number of EFMs or ECMs that are supersets of each of the other pathway definitions. Bars are layered behind each other.

e_coli_core and 45 for iIT341) in uptake and secretion combinations that were generally as expected for growing bacteria (S2 Fig).

Comparing pathway lengths, i.e., the number of reactions participating in each unique growth-supporting pattern, we found that the number of shortest pathways was always the same for all pathway definitions, which led us to compare all pairs of patterns by intersecting them with each other (Fig 3). Indeed, the shortest pathways were identical to each other across definitions. More surprisingly, we found that the ECMs were a superset of the EFPs, which in turn were a superset of the MPs. Specifically, the MPs were equal to the EFPs and ECMs that were support-minimal, i.e., that only included metabolite exchanges required to support growth. Counting the number of MPs that were a subset of each of the other pathways, we found a decreasing trend from one to 11 MP subsets. For ECMs, this trend was consistent between e_coli_core and iIT341 despite the latter model being nearly six times larger than the former with four times as many boundary reactions. Also, all e_coli_core and iIT341 EFPs had exactly one MP subset each. In contrast to this, the number of EFMs and ECMs that were supersets of other pathways ranged from one to about 200 EFMs for e_coli_core and from one to more than 4,300 ECMs for iIT341. Again, we found a trend: for both models, the average number of supersets decreased from EFMs to ECMs to EFPs to MPs. All subset counts as well as EFM and ECM superset counts correlated positively with each other and with pathway length, while superset counts correlated negatively with pathway length for other pathway definitions (S3 and S4 Figs).

For each exchanged metabolite, we computed the frequency of uptake and secretion in all enumerated pathways as well as differences in frequency between pathway definitions (Fig 4). In both e_coli_core and iIT341, a core set of metabolite exchanges was included in all pathways, and thus essential, while most metabolites were only exchanged in one or a few pathways. In general, both individual and pairwise exchange frequencies were consistent across pathway definitions (S5 and S6 Figs). However, the frequencies of some metabolite exchanges differed by more than 10 percentage points between pathway definitions. For e_coli_core,

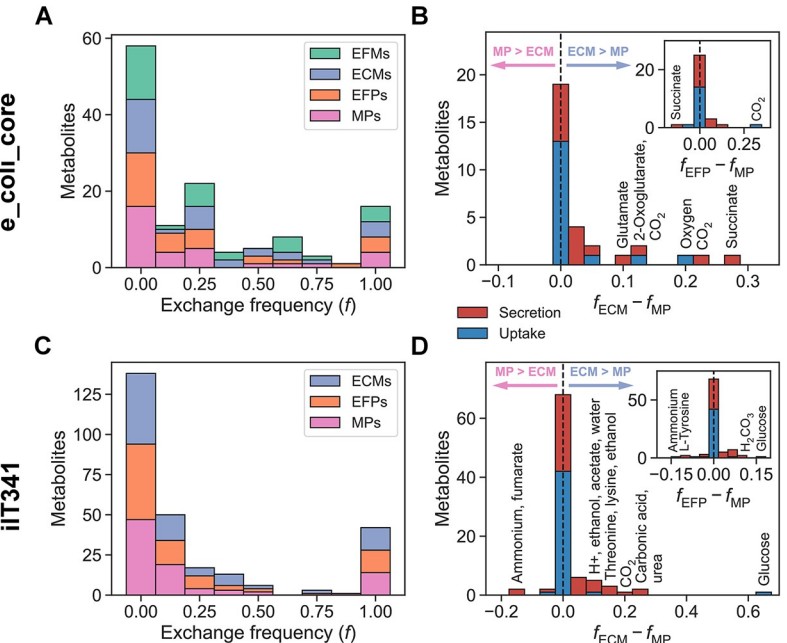

**Fig 4. Metabolite exchange frequencies for microbial species.** (A) Metabolite exchange frequency ($f$), i.e. the fraction of enumerated pathways that included exchange of each metabolite, for e_coli_core. A frequency of zero means a metabolite is never exchanged, while a frequency of one means it is always exchanged. (B) Differences in metabolite exchange frequencies between ECMs and MPs ($f_{ECM} - f_{MP}$) and between EFPs and MPs ($f_{EFP} - f_{ECM}$) for e_coli_core. (C) Metabolite exchange frequency ($f$) in ECMs, EFPs, and MPs for iIT341. (D) Differences in metabolite exchange frequencies between ECMs and MPs ($f_{ECM} - f_{MP}$) and between EFPs and MPs ($f_{EFP} - f_{ECM}$) for iIT341.

uptake of oxygen and $CO_2$ and secretion of succinate, $CO_2$, 2-oxoglutarate, and glutamate were overrepresented in ECMs relative to MPs. Uptake of $CO_2$ was also overrepresented in EFPs relative to MPs, while secretion of succinate was underrepresented. Comparing ECMs and MPs for iIT341, uptake of glucose and $H^+$ and secretion of urea, carbonic acid ($H_2CO_3$), $CO_2$, threonine, lysine, ethanol, acetate, and water were overrepresented in ECMs, while secretion of ammonium and fumarate were underrepresented. Glucose uptake and carbonic acid secretion were overrepresented in EFPs relative to MPs, while ammonium and fumarate secretion were underrepresented.

Applying random flux sampling to boundary reactions in e_coli_core and iIT341, we obtained probability distributions over metabolite exchange fluxes that were consistent between two different samplers (S7–S10 Figs). These probability distributions gave an overview of the growth-supporting flux space, but they could not be directly decomposed to the elementary metabolite conversions provided by pathways. Notably, for e_coli_core, none of the 100,000 samples included $CO_2$ uptake, which was feasible in the model and part of several pathways. By sampling fluxes separately for each of the 34 e_coli_core MPs, we found that $CO_2$ uptake fluxes were indeed accessible for samplers and that sampling could complement pathway analysis by allowing detailed analysis of individual pathways' flux spaces (S11 Fig).

## Metabolite exchanges in a microbial community

To investigate intermicrobial metabolite exchange in a phototrophic mat community during daylight, we built a microbial community model with 132 reactions, 15 of which were boundary reactions, by connecting core models of three individual microbes to each other:

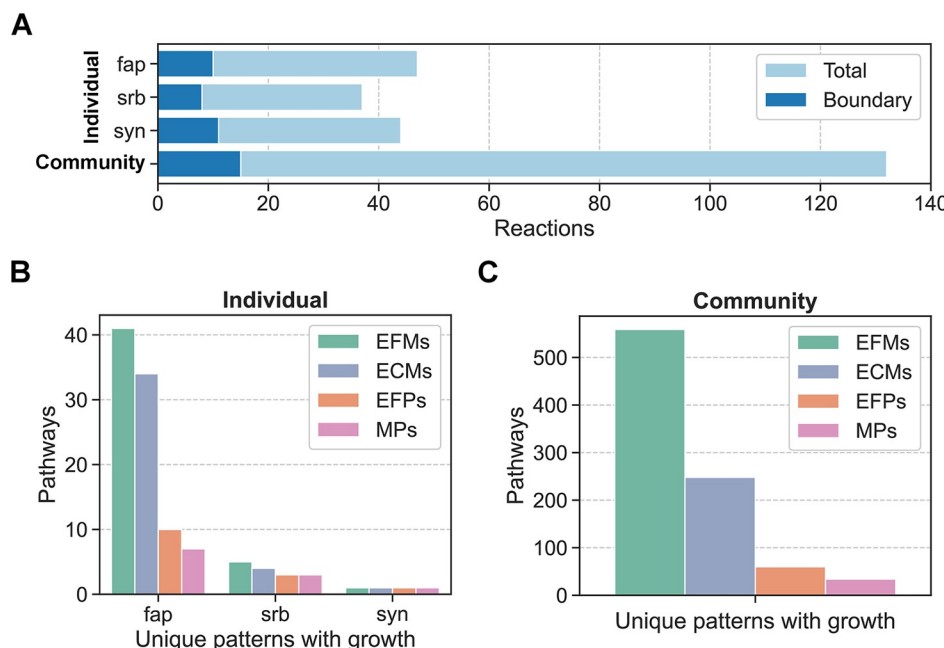

**Fig 5. Number of reactions and pathways for a microbial community.** (A) Number of reactions and boundary reactions in individual and community models. (B) Number of enumerated pathways in individual models (unique patterns with growth). (C) Number of enumerated pathways in community model (unique patterns with growth).

*Synechococcus* spp. (syn) with 37 reactions (8 boundary), a filamentous anoxygenic phototroph (fap) with 47 reactions (10 boundary), and a sulfate-reducing bacterium (srb) with 44 reactions (11 boundary) [26]. We enumerated pathways for the individual models and the community model (Fig 5). Computation times for enumeration ranged from 0.006 s to 4.5 s for the individual models, while enumeration of the community model took 3 min 4 s for EFMs, 2 min 30 s for ECMs, and 0.4 s for MPs (S1 Fig). Pathway counts consistently decreased from fap to srb to syn, and from EFMs to ECMs to EFPs to MPs. We also saw a consistent decrease in counts for the community model pathways, which included both intermicrobial and environmental metabolite exchanges. All pathways included the same metabolites in expected uptake and secretion combinations [26] with the exception of the fap EFMs and ECMs, which included two metabolite exchanges that were not part of any EFPs or MPs (S12 Fig).

Comparing ECMs, EFPs, and MPs, we found that pathways from the individual models were shorter than the community model pathways, and that pathways were sub- and supersets of each other as we saw for microbial species (Fig 6). Also in line with results from microbial species, we found MP subset counts decreasing from one to 12, EFM superset counts ranging from one to 135, and EFM superset counts decreasing, on average, from EFMs to ECMs to EFPs to MPs. We found positive correlation between subset counts for all pathways and between superset counts for EFMs and ECMs, and pathway length correlated positively with subset counts but negatively with superset counts (S13 and S14 Figs).

In both the individual models and the community model, we found a core set of essential metabolite exchanges as well as similar metabolite exchange frequencies across all pathways (Fig 7, S15 and S16 Figs). For the individual models, uptake of acetate, light, and $CO_2$ by fap and secretion of $CO_2$ by srb were more frequent in ECMs than in MPs. Uptake of light, glycolate, and acetate by fap were overrepresented in EFPs relative to MPs, while uptake of oxygen

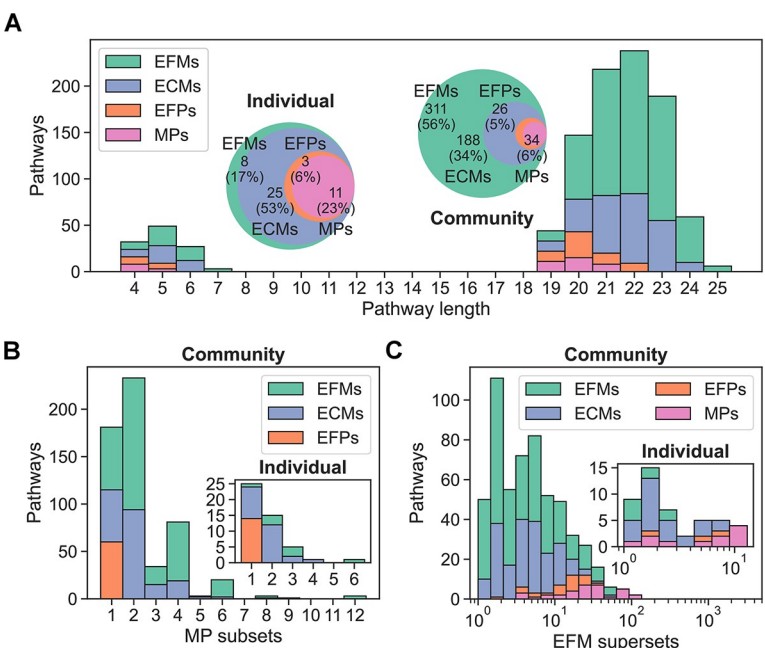

**Fig 6. Comparison of pathways for a microbial community.** (A) Distribution of pathway lengths for individual and community models. Bars are stacked on top of each other. Venn diagrams of pathways are shown with number of unique growth-supporting patterns and percent of the total number of patterns. (B) Distribution of the number of MPs that are subsets of each of the other pathway definitions for individual and community models. Bars are layered behind each other. (C) Distribution of the number of EFMs that are supersets of each of the other pathway definitions for individual and community models. Bars are layered behind each other.

by fap was underrepresented. For the community model, uptake of acetate by fap and srb, secretion of acetate by syn, uptake of light by fap, and uptake of $H_2$ from the environment were overrepresented in ECMs compared to MPs. Uptake of $H_2$ by fap and from the environment were overrepresented in EFPs relative to MPs, while $CO_2$ secretion by fap was underrepresented. As observed for environmental metabolite exchanges in microbial species, the frequencies of intermicrobial metabolic interactions were consistent between pathway definitions. All pathway definitions agreed on all of the following metabolite exchanges: acetate and ammonia from syn to fap and to srb, $CO_2$ from fap to srb and to syn, $CO_2$ from srb to fap and to syn, glycolate from syn to fap, $H_2$ from fap to srb, $H_2$ from srb to fap, and oxygen from syn to fap.

Random flux sampling of the individual and community models produced probability distributions over environmental and intermicrobial metabolite exchange fluxes (S17 and S18 Figs). This showed that some metabolite exchanges were constrained to narrow flux ranges, especially in the community model, while others were free to vary across several orders of magnitude. The flux probability distributions tended to be wider in the individual models than in the community model. For example, ammonia production and secretion fluxes were uniformly distributed across the feasible flux range in the individual syn model, but constrained to a narrow range of small fluxes in the community model.

## A general hierarchical relationship between pathway definitions

The hierarchical relationship between pathway definitions that we found for microbial metabolite exchanges generalizes to arbitrary metabolic networks and subnetworks. To show this, we

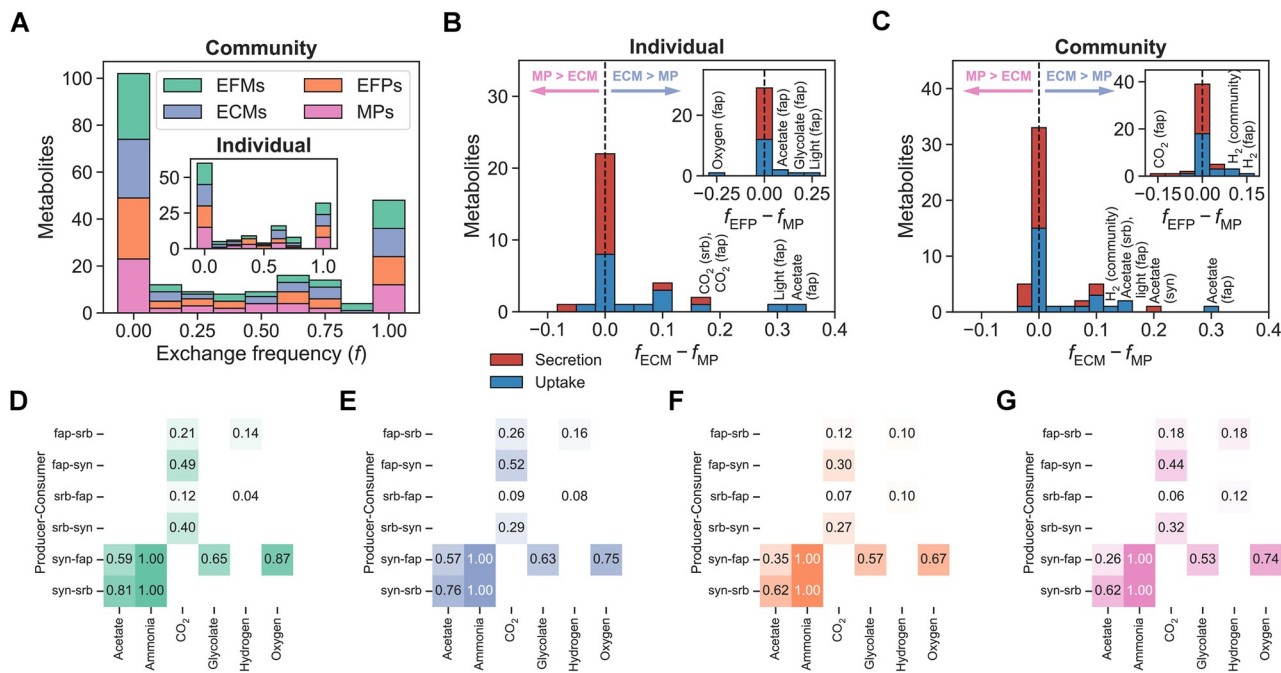

**Fig 7. Metabolite exchange frequencies for a microbial community.** (A) Metabolite exchange frequency ($f$), i.e., the fraction of enumerated pathways that included exchange of each metabolite, in pathways for individual and community models. A frequency of zero means a metabolite is never exchanged, while a frequency of one means it is always exchanged. (B)–(C) Differences in metabolite exchange frequencies between ECMs and MPs ($f_{ECM} - f_{MP}$) and between EFPs and MPs ($f_{EFP} - f_{MP}$) for individual and community models. (D)–(G) Metabolic interactions from (D) EFMs, (E) ECMs, (F) EFPs, and (G) MPs. Rows are producer-consumer pairs, columns are metabolites, and each cell shows a metabolite exchange frequency.

start from a metabolic network $N$ with $m$ metabolites and $n$ reactions, here represented as the set of $n$ reaction indices:

$$N = \{1, 2, \ldots, n\}. \tag{3}$$

Any metabolic network can be formulated such that all reactions are irreversible with positive flux [29], and we assume that this is the case for $N$. We also define a subnetwork of the metabolic network, $S \subseteq N$.

The stoichiometric matrix of $N$ is $\mathbf{N} \in \mathbb{R}^{m \times n}$, constrained by the $m$ metabolite mass balances from Eq 1 and the $n$ flux bounds from Eq 2 with all fluxes required to be positive. Considering only these homogeneous constraints, which are sufficient for our purposes, the feasible flux space of $N$ is an $n$-dimensional flux cone [30],

$$C = \{\mathbf{r} \in \mathbb{R}^n \mid \mathbf{N}\mathbf{r} = 0 \text{ and } \mathbf{r} \geq 0\}, \tag{4}$$

where $\mathbf{r}$ is a flux vector. The flux space of the subnetwork $S$, taking all constraints on $N$ into account, is the projection of $C$ onto the reactions in $S$ [17, 19]. Any linear subspace, such as the flux space of $N$ or its projection onto $S$, is generated without cancellations by a unique minimal set of elementary vectors [31], EFVs in the case of flux spaces [5].

EFMs and ECMs are special cases of EFVs: EFMs can be defined as the elementary vectors of a flux cone [5], and ECMs can be defined as the elementary vectors of the projection of a flux cone onto a subnetwork [17]. Specifically, the EFMs are the EFVs that generate $C$, and the ECMs are the EFVs that generate the projection of $C$ onto $S$. It has been shown that any set of conformal generators of a flux space, i.e., vectors that generate a flux space without

cancellations, is a superset of the elementary vectors of that space [32]. Since the EFMs generate $C$, the parts of the EFMs that are in $S$ generate the projection of $C$ onto $S$ [19]. It follows that the parts of EFMs that are in $S$ are a superset of the ECMs of $S$.

Next, we define a flux pattern as the vector obtained by taking the sign of each element in a flux vector. When all fluxes are required to be positive, this is equivalent to the set of fluxes that are non-zero in the flux vector, i.e., the support of the flux vector. We denote the set of unique flux patterns in $S$ obtained from EFMs, ECMs, EFPs, or MPs as $P_{\mathrm{EFM}}$, $P_{\mathrm{ECM}}$, $P_{\mathrm{EFP}}$, or $P_{\mathrm{MP}}$, respectively. As already shown, the ECMs of $S$ are a subset of the parts of the EFMs that are in $S$, implying that $P_{\mathrm{ECM}} \subseteq P_{\mathrm{EFM}}$. EFPs and MPs are not defined as flux vectors in $N$ or $S$, only as flux patterns in $S$. The EFPs of $S$ are the flux patterns of EFMs from $N$ that correspond to reactions in $S$, but only those patterns that cannot be built from other patterns without cancellations. The MPs are sets of reactions in $S$ that all need to have non-zero flux to satisfy all constraints on $N$, i.e., the support-minimal flux patterns.

The ECM flux patterns of $S$ must include all flux patterns that cannot be built from others without cancellation, i.e., the EFPs of $S$. The reason for this is that they correspond to feasible flux vectors in the subnetwork flux space that the ECMs are required to generate by definition [17, 19]. The flux patterns of ECMs can include patterns that cannot be built from others without cancellation and therefore are not EFPs. However, it is also possible that none of the ECM flux patterns of $S$ can be built from others without cancellation (see example in Fig 1), and in this case the ECM flux patterns are equal to the EFPs. It follows that $P_{\mathrm{EFP}} \subseteq P_{\mathrm{ECM}}$ in general.

Finally, all pathway definitions must include the support-minimal flux patterns of $S$, i.e., the MPs of $S$. These patterns also correspond to EFMs, ECMs, and EFPs, because neither the patterns nor their corresponding flux vectors can be built from other EFMs, ECMs, or EFPs without cancellation. If all EFPs are support-minimal, the EFPs are equal to the MPs, but EFPs are not required to be support-minimal in general. Thus, $P_{\mathrm{MP}} \subseteq P_{\mathrm{EFP}}$. In summary, we have shown that

$$P_{\mathrm{MP}} \subseteq P_{\mathrm{EFP}} \subseteq P_{\mathrm{ECM}} \subseteq P_{\mathrm{EFM}}, \tag{5}$$

i.e., that the flux patterns of pathway definitions are related through a hierarchy for arbitrary metabolic networks and subnetworks. If the EFMs, ECMs, EFPs, and MPs in $S$ are all support-minimal, $P_{\mathrm{MP}} = P_{\mathrm{EFP}} = P_{\mathrm{ECM}} = P_{\mathrm{EFM}}$ (see example in Fig 5).

The hierarchical relationship between pathways in Eq 5 also holds for subsets of pathways that include specific target reactions, e.g., growth as represented by a biomass reaction. The EFMs that include the target reactions will generate the subspace of $C$ where the fluxes of the target reactions are non-zero, and this subspace can be projected onto $S$ just as $C$ can [17, 19]. This projection of the subspace must then be generated by the subset of ECMs that include the target reactions, and the EFPs that include the target reactions must be a subset of these ECMs since the full set of EFPs is a subset of the full set of ECMs. The MPs that include growth must also correspond to the support-minimal growth-supporting flux patterns of EFMs, ECMs, and EFPs.

## Discussion

We have shown that pathway definitions are related through the hierarchical relationship in Eq 5, both for metabolic networks and subnetworks in general and specifically for growth-supporting metabolite exchanges in models of microbial species and communities. To achieve this, we relied on the definitions of EFMs and ECMs as elementary vectors of the flux cone and its projection onto a subnetwork, respectively, as well as an existing proof that any set of conformal generators of a flux space are a superset of its elementary vectors [32]. Comparing

enumerated pathways as flux patterns, we always found that the EFMs were a superset of the ECMs, which were a superset of the EFPs, which were a superset of the MPs. Distributions of pathway counts, pathway lengths, and degree of overlap between pathways consistently reflected this hierarchy, which holds as long as pathways are enumerated under exactly the same conditions, i.e., with equivalent networks, subnetworks, and constraints.

Knowing about the pathway hierarchy makes it easier to interpret enumerated pathways in terms of biology. Notably, all pathway definitions except MPs allow reactions that are not strictly necessary for network functionality to be included in pathways, meaning, for example, that metabolite uptakes or secretions that are not needed for growth can be found in ECMs, EFPs, and EFMs. These exchanges correspond to flux distributions that are non-minimal but feasible, which is in line with observed bacterial secretion of valuable compounds, e.g., in over-flow metabolism [33]. Non-minimal pathways may capture the fact that cells likely need to balance multiple objectives, none of which are perfectly optimized, while maintaining flexibility to respond to perturbations [34]. In general, the pathway hierarchy can help decompose metabolic activities and rank reactions by their contribution to network functionalities.

Pathways found by different methods generally included the same metabolites in similar combinations of uptake and secretion despite large differences in the total number of pathways enumerated. This is illustrated by the distributions of subsets and supersets, which display an increasing degree of overlap with other pathways the more pathways are enumerated. On average, the number of MP subsets increased while the number of EFM or ECM supersets decreased with the number of enumerated pathways. We interpret a pathway's subset counts as the number of other pathways that are needed to summarize it, and its superset counts as the number of other pathways it summarizes. Superset counts were orders of magnitude larger and grew faster with model size than subset counts, which were very well-preserved between models differing in size and structure. The number of EFMs and ECMs grew rapidly with model size, but the many additional pathways tended to summarize few EFPs and MPs. Conversely, EFPs and MPs tended to summarize many of the additional EFMs and ECMs. EFPs always had exactly one MP subset and vice versa, indicating that EFPs and MPs are closely related to each other. Specifically, the EFPs consist of the support-minimal pathways that are shared by all pathway definitions, i.e., the MPs, plus a minimal number of additional pathways with non-minimal supports that are needed to construct all EFMs and ECMs.

Further similarities between pathway definitions can be noted in the individual and pairwise frequencies of metabolite exchanges, which were highly conserved within each model. This shows that different pathway enumeration methods can provide similar predictions of network capabilities and thus allow consistent biological interpretation. In line with this, large differences in pathway counts between methods were explained by small sets of overrepresented metabolites. For example, most of the 123,717 ECMs from iIT341 that were not MPs were accounted for by 11 metabolite uptakes and secretions that were not strictly required for growth. Notably, glucose uptake was about 60 percentage points more frequent in ECMs than in MPs, reflecting the fact that *H. pylori* can synthesize pyruvate from other sources than glucose such as alanine [35]. As MPs are the only pathways required to be support-minimal, they omit metabolites that are not essential for each pathway, explaining why glucose was not frequently featured in MPs compared to ECMs and EFPs. Conversely, secretion of ammonium was more frequent in MPs than in ECMs and EFPs because pathway length is minimized by breaking down as many nitrogenous waste products as possible to ammonium instead of secreting compounds such as urea, threonine, or lysine.

Our results from microbial species also held for microbial communities, which have become a recent focus in constraint-based modeling [36]. In these models, one is often especially interested in analyzing interactions between species and therefore focus on extracellular

activities, providing a natural application for metabolite exchange enumeration. However, microbial community models also differ from models of microbial species in terms of both size and strucure: they consist of multiple single-species models that are combined into the same model and connected through their exchange reactions. This means that the single-species exchange reactions become internal exchange reactions in the microbial community model, which also includes its own external exchange reactions that allow metabolite exchange with the environment. Despite this, we found that distributions of pathway counts, pathway lengths, and degree of overlap between pathways for internal and external exchanges in the community model were very similar to those we saw for external exchanges in the single-species models. We also found the same hierarchy of pathways from EFMs to ECMs to EFPs to MPs. Thus, we found that all pathway definitions included in this study could be applied to a subnetwork that contained both external and internal exchange reactions.

Random flux sampling provides flux probability distributions for metabolite exchanges that can complement enumerated pathways, which we also saw in this study. However, some enumerated pathways did not have a corresponding flux vector obtained by flux sampling. One example of this was uptake of $CO_2$ by *E. coli*, which can indeed fixate $CO_2$ using only native enzymes [37]. One possible explanation is that some subspaces corresponding to specific flux patterns are diminishingly small in comparison to the rest of the solution space, making them hard to access for samplers and requiring a very high number of samples to be detected. As a way to avoid this problem and an example of complementary pathway enumeration and flux sampling, we applied flux sampling separately to each pathway to obtain individual flux probability distributions. These distributions can be weighted by relative pathway usage and combined into a global flux probability distribution, although the true weights will generally be unknown.

Due to increasing pathway counts, enumeration should become progressively harder from MPs to EFPs to ECMs to EFMs. However, as we saw from the computation times recorded in this study, the computational efficiency and scalability of enumeration algorithms and their implementations differ greatly between pathway definitions. EFM and ECM enumeration is based on efficient implementations of the double description method [29] or lexicographic reverse search [38], but still scales poorly because the number of flux vectors grows combinatorially with network size. EFP counts were generally much smaller than EFM and ECM counts and appeared to grow more slowly with network size, but the current implementation of EFP enumeration is based on a mixed-integer linear program (MILP) that includes two binary variables for each reaction in the subnetwork [15]. This MILP scales poorly with subnetwork size and we therefore chose to extract EFPs from enumerated EFMs and ECMs. MP enumeration also involves binary variables, but these are separated from the continuous flux variables into a binary integer program (BIP) that is alternated with multiple linear programs (LPs), which is is more efficient than a MILP [39]. Combined with the minimal number of MPs, this helps make MP enumeration relatively scalable. Ultimately, the choice of pathway definition and enumeration method should be guided by biological questions and the size of the system of interest. For example, if one is interested in detailed analysis of all feasible metabolite exchange fluxes in a small network, one should enumerate EFMs or ECMs. If the patterns of these fluxes are sufficient, EFPs should allow scaling to larger subnetworks. Support-minimal pathways suffice for many key applications [5], and in these cases MPs would be a natural choice.

In summary, our results provide perspectives on how different pathway definitions can be interpreted in relation to each other. Specifically, pathway definitions are related to each other through a hierarchy of flux patterns that generalizes to arbitrary metabolic networks and subnetworks. This helps explain the consistency of our metabolite exchange predictions across models and pathway definitions as well as the relative scalability of pathway enumeration. It

could both aid researchers in choosing methods that best suit their aims and facilitate further development of pathway definitions and enumeration methods. For example, ECMs can be generalized by using the concept of flux cone projection onto a subnetwork instead of defining the conversion cone as a separate solution space. More generally, our findings highlight the potential for further development of a unified framework for pathway analysis, in which all pathways are defined in terms of elementary flux vectors and their patterns in full and projected flux spaces [5, 19]. This could help make unbiased analysis more accessible and scalable, e.g., by combining elements of enumeration methods for different pathway definitions with each other as well as with flux sampling. We hope that this will eventually allow for detailed analysis of metabolic networks across scales, from organisms to communities or even ecosystems [40].

## Methods

### Metabolic models

We obtained and analyzed an *Escherichia coli* core model (e_coli_core) [7], a *Helicobacter pylori* GEM (iIT341) [28], and core models of a filamentous anoxygenic phototroph (fap), a sulfate-reducing bacterium (srb), and *Synechococcus* spp. (syn), both individually and interacting with each other in a microbial community model [26]. The models e_coli_core and iIT341 were downloaded from the *ecmtool* repository (https://github.com/SystemsBioinformatics/ecmtool). We set the bounds of iIT341 to reflect the minII medium [28] and kept the minimal glucose medium for e_coli_core. Using COBRApy [41], we built individual models of fap, srb, and syn by adding the reactions listed for the compartmentalized daylight scenario in the supplementary information of Taffs et al. [26] to the models. We added exchange reactions for all metabolites listed as external and manually identified sink and demand reactions as defined by Thiele et al. [42]. The community model was made by merging the individual models into a new model and connecting them through exchanged metabolites in a shared compartment. Specifically, the exchange reactions of the individual models were connected to the shared metabolites and new exchange reactions between the shared compartment and the environment were created, allowing the intermicrobial and environmental exchanges described in Fig 1A of Taffs et al. [26]. A pseudo-metabolite consisting of equal shares of biomass from each of the individual microbes was constructed and used as the substrate in a community biomass reaction, thus requiring balanced growth of all three microbes.

### EFM enumeration

We used the Python interface of *efmtool* [29] to enumerate EFMs for e_coli_core, fap, srb, syn, and the microbial community model. Irreversible reactions defined with negative flux, i.e., from right to left, were reversed before enumeration. We extracted unique exchange patterns from EFMs by removing internal reactions, taking the sign of the reduced EFM matrix, and removing duplicate patterns. For the community model, we included both environmental and intermicrobial exchanges. EFM enumerations were performed on a Lenovo ThinkPad laptop with an Intel Core i7-8665U (1.90GHz) processor and 32 GB RAM.

### ECM enumeration

We used the Python package *ecmtool* to enumerate ECMs [18]. To reproduce the results of [18], we ran ECM enumerations for e_coli_core and iIT341 using the command-line interface of *ecmtool* and scripts based on their supplementary information. These ECM enumerations were performed on the Orion cluster at the Norwegian University of Life Sciences (NMBU),

using one core for e_coli_core and four cores for iIT341. We enumerated ECMs for fap, srb, syn, and the microbial community model using the Python interface of *ecmtool* with default settings. For the community model, we included both environmental and intermicrobial exchanges by using the "tag" method of *ecmtool*. We converted ECMs to unique exchange patterns by taking the sign of the ECM matrix and removing duplicate patterns. These ECM enumerations were performed on a Lenovo ThinkPad laptop with an Intel Core i7-8665U (1.90GHz) processor and 32 GB RAM.

### EFP enumeration

We enumerated EFPs from the unique exchange patterns of EFMs or ECMs. Specifically, we identified EFPs as the patterns that could not be constructed from other patterns without cancellations. For each pattern, we took the union of all other patterns of which it was a superset and checked whether this union was equal to the pattern itself.

### MP enumeration

We used the Python package *mptool* [20] to enumerate MPs. Specifically, we used the iterative graph method and default settings. We added a minimal growth rate requirement by setting the lower bound of the biomass reaction to a small positive value ($10^{-4}$ $h^{-1}$). All reversible boundary reactions were split into two irreversible reactions to distinguish production from consumption of metabolites. The subset of irreversible boundary reactions was then chosen as the subnetwork for MP enumeration. For the microbial community model, we also included intermicrobial metabolite exchanges in the subnetwork. All MP enumerations were performed on a Lenovo ThinkPad laptop with an Intel Core i7-8665U (1.90GHz) processor and 32 GB RAM.

### Comparing pathways

EFMs and ECMs were enumerated as vectors, in which each element is a stoichiometric coefficient or flux, while EFPs and MPs were enumerated as unique patterns that only include the signs of these elements. MPs must also satisfy a functional requirement, in our case a minimal growth rate. To make all pathways comparable, we therefore extracted unique exchange patterns from EFMs and ECMs and further extracted growth-supporting patterns for EFMs, ECMs, and EFPs. We converted the patterns to sets, distinguishing between import and export, and intersected all pairs of pathways to count subsets and supersets (see example in Fig 1).

### Metabolite exchange frequencies

We computed metabolite exchange frequencies separately for EFMs, ECMs, EFPs, and MPs by counting the number of pathways in which each metabolite was exchanged and dividing by the total number of pathways. Pairwise metabolite exchange frequencies were computed in the same way, i.e., by counting the number of pathways in which each pair of metabolites was exchanged together and dividing by the total number of pathways. For the microbial community model, we also computed the frequency of metabolic interactions, i.e., intermicrobial metabolite exchanges. We counted the number of pathways in which each metabolite was produced by each microbe and consumed by each other microbe and divided by the total number of pathways, separately for EFMs, ECMs, EFPs, and MPs.

### Random flux sampling

Uniform random flux sampling of e_coli_core and iIT341 was performed using the probabilistic thermodynamic analysis (PTA) flux sampler (without considering thermodynamics) from the Python package *pta* [43]. For each model, we set the lower bound of the biomass reaction to $0.1 \, h^{-1}$ and sampled 100,000 flux vectors from the resulting solution space. We ensured the space was sufficiently sampled by checking the convergence of the sampler with the "check_convergence" method of *pta*. These samplings were performed on a Lenovo ThinkPad laptop with an Intel Core i7-1165G7 (2.80GHz) prosessor and 16 GB RAM. We also applied the OptGP sampler [44] from COBRApy [41], to the same models with very similar results. Subsequently, we obtained 1,000 samples for each enumerated MP from e_coli_core and 100,000 samples each for fap, srb, syn, and the microbial community model using OptGP. These samplings were performed on a Lenovo ThinkPad laptop with an Intel Core i7-8665U (1.90GHz) processor and 32 GB RAM. We used flux variability analysis (FVA) to ensure that samples were within the feasible flux ranges [45].

## Supporting information

**S1 Fig. Computation times for pathway enumeration.** Computation time for enumerating all EFMs, ECMs, or MPs for all models analyzed in this study. Models used to analyze microbial species and models used to analyze a microbial community are separated by a dashed line. EFPs were extracted from EFMs or ECMs and therefore not enumerated separately. All enumerations except ECM enumeration for iIT341 were performed on the same laptop computer (see Methods for details).
(TIFF)

**S2 Fig. Clustered heatmaps of pathways for microbial species.** Clustered heatmaps of (A) EFMs for e_coli_core, (B) ECMs for e_coli_core, (C) EFPs for e_coli_core, (D) MPs for e_coli_core, (E) ECMs for iIT341, (F) EFPs for iIT341, and (G) MPs for iIT341. Rows are metabolites, columns are pathways, and each cell indicates metabolite uptake (blue) or secretion (red) in a pathway. Rows and columns are clustered by Ward's minimum variance method. Unique growth-supporting flux patterns are shown.
(TIFF)

**S3 Fig. Pairwise relationships between pathway lengths and counts for e_coli_core.** Pairwise relationships between (A) pathway length and number of EFM, ECM, EFP, and MP subsets and (B) pathway length and number of EFM, ECM, and EFP supersets for e_coli_core.
(TIFF)

**S4 Fig. Pairwise relationships between pathway lengths and counts for iIT341.** Pairwise relationships between (A) pathway length and number of ECM, EFP, and MP subsets and (B) pathway length and number of ECM and EFP supersets for iIT341.
(TIFF)

**S5 Fig. Metabolite exchange frequencies for microbial species.** Metabolite exchange frequencies (fraction of pathways including secretion or uptake of each metabolite) for (A) e_coli_core and (B) iIT341.
(TIFF)

**S6 Fig. Pairwise metabolite exchange frequencies for microbial species.** Pairwise metabolite exchange frequencies (fraction of pathways including secretion or uptake of each metabolite pair) for (A) EFMs for e_coli_core, (B) ECMs for e_coli_core, (C) EFPs for e_coli_core, (D) MPs for e_coli_core, (E) ECMs for iIT341, (F) EFPs for iIT341, and (G) MPs for iIT341. Rows

and columns are metabolites and each cell indicates the pairwise exchange frequency of two metabolites.
(TIFF)

**S7 Fig. Flux probability distributions from PTA for metabolite exchanges in e_coli_core.** Flux probability distributions for metabolite exchanges in e_coli_core from 100,000 random flux vectors sampled with PTA. Dashed lines indicate feasible flux ranges from FVA.
(TIFF)

**S8 Fig. Flux probability distributions from OptGP for metabolite exchanges in e_coli_core.** Flux probability distributions for metabolite exchanges in e_coli_core from 100,000 random flux vectors sampled with OptGP. Dashed lines indicate feasible flux ranges from FVA.
(TIFF)

**S9 Fig. Flux probability distributions from PTA for metabolite exchanges in iIT341.** Flux probability distributions for metabolite exchanges in iIT341 from 100,000 random flux vectors sampled with PTA. Dashed lines indicate feasible flux ranges from FVA.
(TIFF)

**S10 Fig. Flux probability distributions from OptGP for metabolite exchanges in iIT341.** Flux probability distributions for metabolite exchanges in iIT341 from 100,000 random flux vectors sampled with OptGP. Dashed lines indicate feasible flux ranges from FVA.
(TIFF)

**S11 Fig. Flux probability distributions from OptGP for metabolite exchanges in each MP from e_coli_core.** Flux probability distributions for metabolite exchanges in e_coli_core from 1,000 random flux vectors sampled with OptGP for each MP. Rows are MPs and columns are metabolites. Dashed lines indicate feasible flux ranges from FVA.
(TIFF)

**S12 Fig. Clustered heatmaps of pathways for a microbial community.** Clustered heatmaps of (A) EFMs for fap, (B) ECMs for fap, (C) EFPs for fap, (D) MPs for fap, (E) EFMs for srb, (F) ECMs for srb, (G) EFPs for srb, (H) MPs for srb, (I) EFMs for syn, (J) ECMs for syn, (K) EFPs for syn, and (L) MPs for syn. Rows are metabolites, columns are pathways, and each cell indicates metabolite uptake (blue) or secretion (red) in a pathway. Rows and columns are clustered by Ward's minimum variance method and rows are colored to indicate individual microbes and the community (yellow). Unique growth-supporting flux patterns are shown.
(TIFF)

**S13 Fig. Pairwise relationships between pathway lengths and counts for individual models.** Pairwise relationships between (A) pathway length and number of EFM, ECM, EFP, and MP subsets and (B) pathway length and number of EFM, ECM, and EFP supersets for fap, srb, and syn individually.
(TIFF)

**S14 Fig. Pairwise relationships between pathway lengths and counts for community model.** Pairwise relationships between (A) pathway length and number of EFM, ECM, EFP, and MP subsets and (B) pathway length and number of EFM, ECM, EFP, and MP supersets for the microbial community model.
(TIFF)

**S15 Fig. Metabolite exchange frequencies for a microbial community.** Metabolite exchange frequencies (fraction of pathways including secretion or uptake of each metabolite) for (A) fap,

(B) srb, (C) syn, and (D) the microbial community model.
(TIFF)

**S16 Fig. Pairwise metabolite exchange frequencies for a microbial community.** Pairwise metabolite exchange frequencies (fraction of pathways including secretion or uptake of each metabolite pair) for (A) EFMs for fap, (B) ECMs for fap, (C) EFPs for fap, (D) MPs for fap, (E) EFMs for srb, (F) ECMs for srb, (G) EFPs for srb, (H) MPs for srb, and (I)–(L) EFMs, ECMs, EFPs, and MPs for the microbial community model. Rows and columns are metabolites and each cell indicates the pairwise exchange frequency of two metabolites.
(TIFF)

**S17 Fig. Flux probability distributions from OptGP for metabolite exchanges in individual models.** Flux probability distributions for metabolite exchanges in fap, srb, and syn individually from 100,000 random flux vectors sampled with OptGP. Dashed lines indicate feasible flux ranges from FVA.
(TIFF)

**S18 Fig. Flux probability distributions from OptGP for metabolite exchanges in community model.** Flux probability distributions for metabolite exchanges in the microbial community model from 100,000 random flux vectors sampled with OptGP. Dashed lines indicate feasible flux ranges from FVA.
(TIFF)

## Acknowledgments

We thank Tom Clement, Daan de Groot, Bianca Buchner, and Jürgen Zanghellini for discussions and help with *ecmtool*, Bas Teusink and the rest of the Systems Biology lab at Vrije Universiteit Amsterdam for hosting us for research stays, Teshome Dagne Mulugeta for helping us set up software on the Orion cluster, Axel Theorell for help with random sampling, and Ross Carlson for help with the microbial community model.

## Author Contributions

**Conceptualization:** Ove Øyås.

**Data curation:** Ylva Katarina Wedmark, Ove Øyås.

**Formal analysis:** Ylva Katarina Wedmark, Jon Olav Vik, Ove Øyås.

**Funding acquisition:** Jon Olav Vik.

**Investigation:** Ylva Katarina Wedmark, Jon Olav Vik, Ove Øyås.

**Methodology:** Ylva Katarina Wedmark, Ove Øyås.

**Project administration:** Ove Øyås.

**Resources:** Ove Øyås.

**Software:** Ylva Katarina Wedmark, Jon Olav Vik, Ove Øyås.

**Supervision:** Jon Olav Vik, Ove Øyås.

**Validation:** Ylva Katarina Wedmark, Ove Øyås.

**Visualization:** Ylva Katarina Wedmark, Ove Øyås.

**Writing – original draft:** Ylva Katarina Wedmark, Jon Olav Vik, Ove Øyås.

**Writing – review & editing:** Ylva Katarina Wedmark, Jon Olav Vik, Ove Øyås.

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
