## [Decision Letter · Decision Letter 0]

31 May 2024

Dear Dr. Øyås,

Thank you very much for submitting your manuscript "A hierarchy of metabolite exchanges in metabolic models of microbial species and communities" for consideration at PLOS Computational Biology.

As with all papers reviewed by the journal, your manuscript was reviewed by members of the editorial board and by several independent reviewers. In light of the reviews (below this email), we would like to invite the resubmission of a significantly-revised version that takes into account the reviewers' comments. In addition, due to the highly technical nature of the study, reviewer #3 suggests that the manuscript be partially rewritten to aid readers in keeping track of the various pathway formulations. Often, such highly technical reports are further obscured by the over use of abbreviations. I ask that you consider these concerns carefully while using your best judgement.

We cannot make any decision about publication until we have seen the revised manuscript and your response to the reviewers' comments. Your revised manuscript is also likely to be sent to reviewers for further evaluation.

Sincerely,

William Cannon

Academic Editor

PLOS Computational Biology

Mark Alber

Section Editor

PLOS Computational Biology

Reviewer's Responses to Questions

**Comments to the Authors:**

Reviewer #1: This study analyzes hierarchical relationships among four types of pathways for genome-scale metabolic models (GEMs) from the perspective of metabolic exchanges. The pathways examined include elementary flux modes (EFMs), elementary flax patterns (EFPs), elementary conversion modes (ECMs), and minimal pathways (MPs). The study demonstrated the general hierarchy as

$P_{MP} \\subseteq P_{EFP} \\subseteq P_{ECM} \\subseteq P_{EFM}$.

They conducted computational experiments using small GEMs, e_coli_core and iIT341 as well as microbial community model based on e_coli_core. These experiments supported the efficiency of focusing on metabolic exchanges.

The idea of focusing on metabolic exchanges and improving the scalability of pathway analysis is reasonable and efficient. However, the derived hierarchy seems obvious from the definitions.

Below are points that seem to require additional explanation.

1.While the idea of ECMs and some other types of pathways is to ignore the internal reactions, lengths are defined for them. The definitions of the length for them should be clarified.

2.Metabolite exchange frequency should be formally defined.

3.Information regarding computation time appears to be lacking. Is it possible to enumerate the pathways focusing on metabolic exchanges for larger GEMs?

Reviewer #2: This study investigated and compared different pathway analysis methods (elementary flux modes, elementary conversion modes, elementary flux patterns, and minimal pathways), using metabolic models of microbial species from small size to genome-scale, as well as community level model of a microbial consortium. The authors mainly analyzed metabolite exchanges between species, or between species and environment, and found that different pathway definitions for metabolic networks or subnetworks were related through a hierarchy. Thank you for doing this work and adding to our knowledge base in this field. Here are some comments for further refinement of the manuscript.

1.) The authors should introduce in more detail about reconstruction process of the microbial community model, e.g. what modeling method was used, how to construct biomass reaction…

2.) For srb model, does it have 209 EFMs (line 140)?

Reviewer #3: Overall comment:

The manuscript is highly technical, but an important study in the field. The hierarchical relationship between pathway definitions is well-explained and supported by extensive computational analysis. I confess I had to read several sections multiple times and had a “cheat sheet” with the definitions of EFMs, ECMs, EFPs on the side since each abbreviation is mentioned close to 100 times each. I understand this is the result of the highly technical nature of this manuscript and while it’s not a dealbreaker for me, any improvements to the readability of the manuscript would be welcome. In particular, consider adding more descriptive captions to the figures across the manuscript, In particular figures 3 and 4 can use caption improvements the most. The amount of data provided in supplementary materials is commendable and allows for a deep dive into how these studies were performed. I also appreciate seeing the clear statement of the author's contributions with a lot of detail. I recommend that this manuscript be accepted for publication with only minor comments below.

Detailed comments:

Abstract:

The abstract provides a concise synopsis of the study question, methodology, findings, and conclusion is given in the abstract. The main conclusions are emphasized along with their importance. However, can the authors add a brief explanation of the study's more general implications?

Introduction:

Lines 4-5: "The growing availability of genomes and other omics data has enabled metabolic network reconstruction in silico, giving rise to genome-scale metabolic models (GEMs) that are usually formulated as constraint-based models (CBMs) to allow scaling." A reference to a review paper on the history and development of GEMs and CBMs would be beneficial here.

In Figure 1, make5 sure that all abbreviations are defined in the caption.

Overall, the manuscript introduction does a good job of explaining pathway definitions and their hierarchical relationships clearly. However, can the authors provide more background on the practical implications of these findings for metabolic engineering and synthetic biology?

Results and Discussion:

Lines 113-116: "For each exchanged metabolite, we computed the frequency of uptake and secretion in EFMs, ECMs, EFPs, and MPs as well as differences in frequency between pathway definitions." Can teh authors provide further detail on how the frequency computation is performed?

Line 332, correct the typo in "lexicograhic" to "lexicographic."

According to the study, every metabolic network or subnetwork can be generalized to have a hierarchical link between pathway descriptions. What are the potential exceptions and/or limitations? Any alternative interpretations?

Can you discuss how the hierarchical relationship between pathway definitions can/will impact future research in the field of metabolic network analysis?

The results of the computation studies are compelling, but can the authors empathize with the biological significance of some conclusions? For example, what new insights (if any) do these studies provide into how microbial communities in industrial settings or natural environments operate?

The manuscript addresses the scalability of various enumeration methods and even mentions that in some cases you could not compute the full set for a given model, but a more thorough description of their computing effectiveness and usefulness for large-scale models would be beneficial.

**Have the authors made all data and (if applicable) computational code underlying the findings in their manuscript fully available?**

Reviewer #1: Yes

Reviewer #2: Yes

Reviewer #3: Yes

PLOS authors have the option to publish the peer review history of their article (what does this mean?). If published, this will include your full peer review and any attached files.

Reviewer #1: No

Reviewer #2: **Yes: **Peishun Li

Reviewer #3: No
---

## [Decision Letter · Decision Letter 1]

9 Sep 2024

Dear Dr. Øyås,

We are pleased to inform you that your manuscript 'A hierarchy of metabolite exchanges in metabolic models of microbial species and communities' has been provisionally accepted for publication in PLOS Computational Biology.

Best regards,

William Cannon

Academic Editor

PLOS Computational Biology

Marc Birtwistle

Section Editor

PLOS Computational Biology

Reviewer's Responses to Questions

**Comments to the Authors:**

Reviewer #1: All my concerns have been appropriately addressed.

Reviewer #2: The authors answered all my questions, so i have no further comments

Reviewer #3: Thank you for addressing all my questions, particularly regarding readability of the manuscript. I have no additional concerns.

**Have the authors made all data and (if applicable) computational code underlying the findings in their manuscript fully available?**

Reviewer #1: Yes

Reviewer #2: Yes

Reviewer #3: None

PLOS authors have the option to publish the peer review history of their article (what does this mean?). If published, this will include your full peer review and any attached files.

Reviewer #1: No

Reviewer #2: No

Reviewer #3: **Yes: **José P. Faria

---

## [Editor Report · Acceptance letter]

17 Sep 2024

PCOMPBIOL-D-24-00266R1 

A hierarchy of metabolite exchanges in metabolic models of microbial species and communities

Dear Dr Øyås,

I am pleased to inform you that your manuscript has been formally accepted for publication in PLOS Computational Biology. Your manuscript is now with our production department and you will be notified of the publication date in due course.

With kind regards,

Jazmin Toth
